

# Impact of land reclamation and agricultural water regime on the distribution and conservation status of the endangered *Dryophytes suweonensis*

Amaël Borzée[1,2], Kyungmin Kim[2], Kyongman Heo[2,3], Piotr G. Jablonski[1,4] and Yikweon Jang[2,5]

[1] Laboratory of Behavioral Ecology and Evolution, School of Biological Sciences, Seoul National University, Seoul, South Korea
[2] Division of EcoScience, Ewha Women's University, Seoul, South Korea
[3] College of Natural Science, Sangmyung University, Seoul, South Korea
[4] Museum and Institute of Zoology, Polish Academy of Sciences, Warsaw, Poland
[5] Interdisciplinary Program of EcoCreative, Ewha Women's University, Seoul, South Korea

## ABSTRACT

Knowledge about the distribution and habitat preferences of a species is critical for its conservation. The Suweon Treefrog (*Dryophytes suweonensis*) is an endangered species endemic to the Republic of Korea. We conducted surveys from 2014 to 2016 at 890 potentially suitable sites across the entire range of the species in South Korea. We then assessed whether *D. suweonensis* was found in the current and ancestral predicted ranges, reclaimed and protected areas, and how the presence of agricultural floodwater affected its occurrence. Our results describe a 120 km increase in the southernmost known distribution of the species, and the absence of the species at lower latitudes. We then demonstrate a putative constriction on the species ancestral range due to urban encroachment, and provide evidence for a significant increase in its coastal range due to the colonisation of reclaimed land by the species. In addition, we demonstrate that *D. suweonensis* is present in rice fields that are flooded with water originating from rivers as opposed to being present in rice fields that are irrigated from underground water. Finally, the non-overlap of protected areas and the occurrence of the species shows that only the edge of a single site where *D. suweonensis* occurs is legally protected. Based on our results and the literature, we suggest the design of a site fitting all the ecological requirements of the species, and suggest the use of such sites to prevent further erosion in the range of *D. suweonensis*.

## INTRODUCTION

Very few species have a cosmopolitan distribution, and most are likely to be under local environmental pressure (*Purvis et al., 2000*). When the entire range of a species is threatened by urbanization or other types of habitat modification, the risk of extinction increases exponentially (*Huxley, 2013*). As a result, the assessment of extinction risks

Corresponding authors
Amaël Borzée,
amaelborzee@gmail.com
Piotr G. Jablonski,
piotrjab@hotmail.com
Yikweon Jang, jangy@ewha.ac.kr

depend on threat levels (*Mace & Lande, 1991*; see *IUCN, 2016*), which may guide optimal conservation effort to prevent extinction (*Pimm et al., 2014*).

Lack of knowledge of species' distributions has already resulted in extinctions that could have been easily avoided. For example, the Tecopa pupfish (*Cyprinodon nevadensis calidae*) became extinct following construction of man-made structures on the Tecopa Hot Springs, the only site where the species occurred (*Miller, Williams & Williams, 1989*). Unfortunately, this information was not available at the time of construction. Knowledge of species' habitat preferences provides background information for the assessment of extinction risks (*Manne & Pimm, 2001*), and can be used to develop spatial models for species' distribution (*Corsi, De Leeuw & Skidmore, 2000*). For instance, a subspecies of Ursini's viper, *Vipera ursinii graeca*, was known to occur only in Greece and at a single locality in Albania. However, eight new localities were found through landscape and climate modelling, doubling the known range of the species (*Mizsei et al., 2016*).

Although critical, obtaining information about species' ranges and habitat preferences is only a first step for any conservation effort. At risk species with clearly defined ranges still go extinct in large numbers and a way to stem this loss is through the implementation of protected areas (*Pimm et al., 2014*). The occurrence of a species within a protected area will significantly increase its chance of survival, despite the debated effectiveness of currently located protected areas (*Abellán & Sánchez-Fernández, 2015*), and the need for the establishment of additional protected areas (*Brooks et al., 2004*).

The Class Amphibia is currently the most endangered class of vertebrates (*Stuart et al., 2004*). Among the difficulties for amphibian conservation efforts are unknown distribution limits and the absence of adequate breeding sites. Suitable natural wetlands for amphibians have been converted into farmlands such as rice-paddies over the last century, especially in the Republic of Korea (*Juliano, 1993*; *Czech & Parsons, 2002*; *Machado & Maltchik, 2010*). Furthermore, those farmlands still holding a fraction of the original biodiversity are being converted into residential and commercial facilities at an alarming rate. In the Republic of Korea, rice production has decreased by about 25% since peak production in the 1970s (*FAO, 2016*). Since then, there have been clear negative repercussions on habitats available for amphibians (*Park et al., 2014*).

The Suweon Treefrog, *Dryophytes suweonensis* (previously attributed to *Hyla*; *Duellman, Marion & Hedges, 2016*), is an endangered, endemic treefrog species from the Korean Peninsula. As of 2012, the species was known to occur in a very restricted range, limited to five valleys centred in metropolitan Seoul (*Kim et al., 2012*). It is therefore possible that the largest populations of *D. suweonensis* might have been historically present in and around the present Seoul area (*Borzée et al., 2015*). Yet, opportunistic observations of calling males in the Democratic People's Republic of Korea (*Chun et al., 2012*) and further south than previously reported (*Borzée, Yu & Jang, 2016*) have lead to the expectation of a broader distribution for the species.

*Dryophytes suweonensis* is an evolutionary important species due to its unusual ZW karyotype, warranting special conservation efforts (*Dufresnes et al., 2015*). Here, we first aimed to describe the extent of occurrence and distribution of the species through occurrence surveys, as well as the loss of ancestral range because of urbanisation. We

then assessed the overlap between the range of the species and reclaimed tidal flats, and the overlap between range and protected areas. Next, because the distribution of *D. suweonensis* is closely intertwined with rice cultivation, we examined whether the origin of agricultural flood waters was critical for the occurrence of the species. Finally, we extracted environmental variables collected from field surveys and described optimal conservation sites for *D. suweonensis*.

## MATERIAL AND METHODS

Field surveys were conducted during 2014, 2015 and 2016, only after the beginning of the breeding season of the species (*Roh, Borzée & Jang, 2014*) to prevent any false negative. Because *Dryophytes suweonensis* has not been observed using other vegetation than rice seedlings as supports from which to hang to produce advertisement calls (*Borzée, Kim & Jang, 2016*), and because it is not known to breed in any other wetland than rice paddies (*Borzée & Jang, 2015*), the species typically starts breeding after rice planting.

The setting of modern rice fields during the last decades led to a specific geometric grouping of rice paddies, here referred to as rice-paddy complexes. A rice-paddy complex is characterized by a central ditch running mostly straight through the complex for irrigation purposes. Along this central ditch, and thus along the longest and straightest line available, usually runs a cemented lane, typically following the centre of the valley. In this study, rice-paddy complexes were considered spatially independent if further than 200 m apart, the maximum daily dispersion distance for the species (*Borzée et al., 2016*), or separated by landscape barriers impermeable to treefrogs (*Roh, Borzée & Jang, 2014*).

The Japanese Treefrog, *D. japonicus* is ubiquitously present on the wetlands of the Korean Peninsula, and the two treefrog species are in sympatry at all sites. The advertisement calls of *D. japonicus* and *D. suweonensis* are species specific (*Jang et al., 2011*; *Park, Jeong & Jang, 2013*), and we noted the presence or absence of *D. suweonensis* through acoustic monitoring. In calling anurans, including Hylids, acoustic monitoring is known to be reliable to estimate population size, and thus adequate to assess occurrence (*Weir et al., 2005*; *Pellet, Helfer & Yannic, 2007*; *Dorcas et al., 2009*; *Petitot et al., 2014*; *Moreira, Moura & Maltchik, 2016*). In a preliminary study, our aural survey protocol with 5-min transects was accurate to estimate the occurrence of *D. suweonensis* (*Borzée et al., 2017*).

### Transect surveys

We defined the general area for this study *a priori*, following the ecological requirements of the species such as defined by *Roh, Borzée & Jang (2014)* and including all natural and man-made wetlands west of 127.5°E and below 120 m above sea level. This pre-selection of potential breeding sites through Google Earth Pro (Google Earth imagery, v7.1.2.2041, 2013) identified 789 sites in 2014 (Fig. 1). A previous study for the occurrence of this species had drawn the southern limit of the range around the Bay of Asan, below 37°N (*Kim et al., 2012*; Fig. 1). However, our surveys in 2014 demonstrated the southern limit of the range to be inaccurate (*Borzée, Yu & Jang, 2016*), and additional surveys were conducted further south in 2015 and 2016, until reaching a point past where the species was no longer detected. In 2015, we surveyed 189 sites, composed of 90 new sites and 99 sites where the

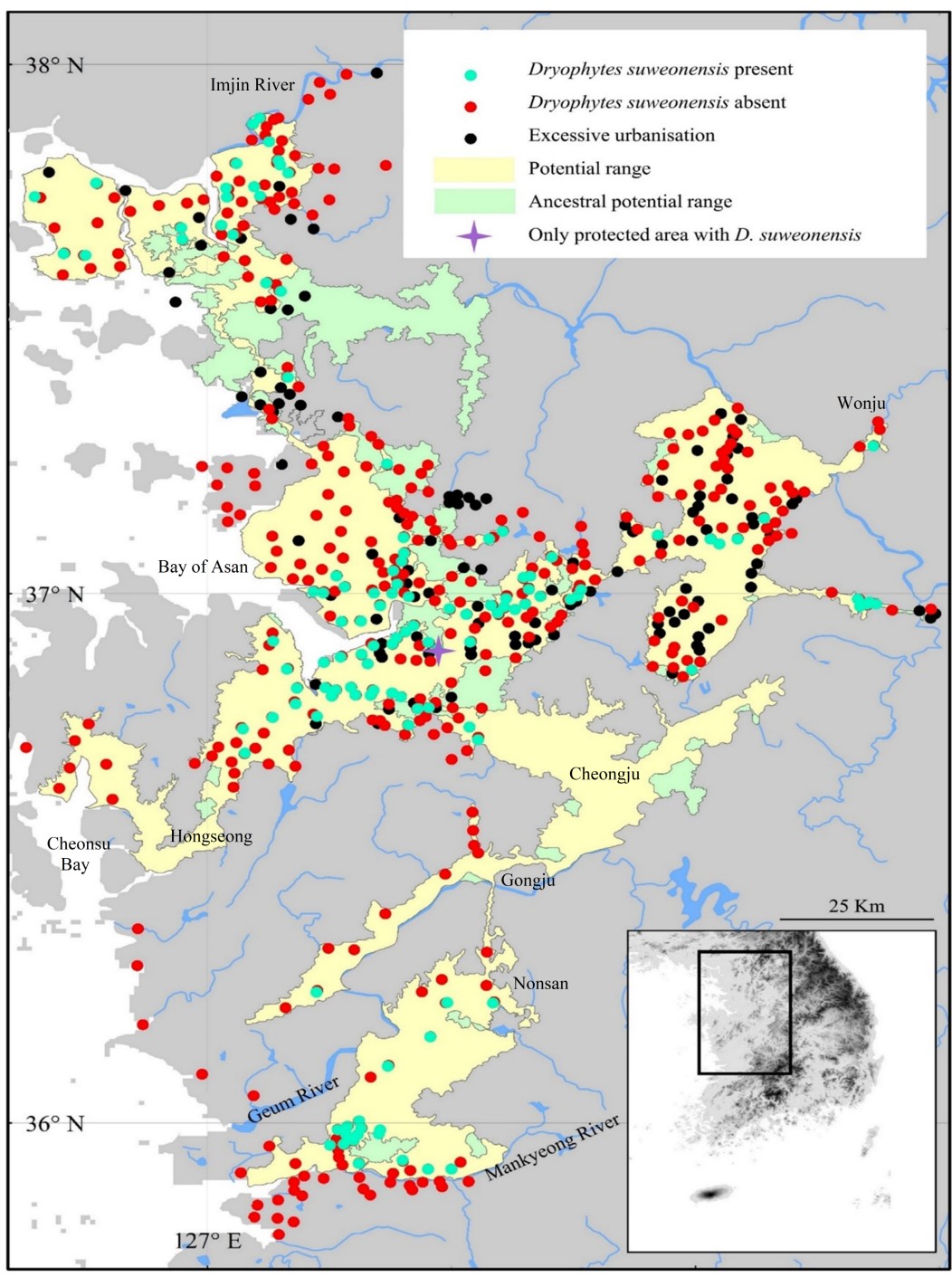

**Figure 1  Summary of the 890 sites surveyed at least once over the three years of surveys.** *Dryophytes suweonensis* was detected at least once at 114 sites, and 421 sites were too excessively urbanised for the species to occur. Here, potential current range is defined as the range where the species could currently occur, while the potential ancestral range is the range where the species could have occurred before urban development. Pyeongtaek is the area where the only protected area with *D. suweonensis* is found.

species was present in 2014. A single site where the species was detected in 2014 could not be visited again due to its location within the Civilian Control Zone (CCZ) adjacent to the border with the Democratic Republic of Korea and the lack of permits for 2015 and 2016. In 2016, we surveyed a total of 122 sites (99 sites from 2014, 12 from 2015 and 11 new sites). All accessible sites where the species had been recorded in 2014 were surveyed in 2015 and 2016, even if the species was not detected in 2015. All sites where the species had been detected in 2015 were kept in the list of sites to survey in 2016. In total, 890 sites were surveyed at least once over the three years of surveys.

Surveys were conducted between 5 pm and 2 am, during the peak calling activity of the species. After arrival at a survey site, five minutes were spent waiting quietly. For each site, aural monitoring was conducted along a single transect along the centre of the rice-paddy complex. A surveyor walked briskly at a maximum speed of *circa* 80 m/min along the transect, noting the presence or absence of *D. suweonensis* at the rice-paddy complex. Before conducting the project, we had empirically measured the detection range for advertisement calls of *D. suweonensis* ($n = 20$), resulting in a $250 \pm 45$ m range. The farthest rice paddies in rice-paddy complexes were typically within this detection range.

At the end of each transect survey we recorded water pH and water conductivity (μS) to define the ecological preferences of *D. suweonensis*. We also estimated surface area and longest straight line within sites to determine a sphericity ratio for the occurrence of the species. This is important for determining the likelihood of a species' presence because a circular site will better retain a species than a narrow and linear site. We then recorded the length of continuity with rivers and forests, defined as the continuous line between the edge of rice-paddy complexes and the aforementioned landscape feature, and finally, we noted the presence of buildings and greenhouses within the rice-paddy complexes. These variables were collected through the drawing of polygons or visual inspection of sites in Google Earth Pro (Google Earth imagery, v7.1.2.2041, 2016), at a 10 m resolution, on map dated from 2015 at the latest.

## Reclaimed lands and protected area

To correlate the presence of the species with shifting landscape use, we recorded the presence of the *D. suweonensis* at sites located on reclaimed lands. Here, reclaimed lands used to be mudflats and sea beds, which have been converted into rice-paddy complexes. To record the presence of reclaimed lands, we compared maps from 1950–51 drawn by the US Army (*Center of Military History, 1990*) downloaded in Google Earth and present satellite pictures from Google Earth Pro (Google Earth imagery, 6.2.2.6613, 2016). The 1950–51 maps were selected due to their precision. A land was considered reclaimed if it was not usable for breeding by *D. suweonensis* in 1950–51, but converted into rice paddies before 2016.

We then compared the presence of protected areas and the localities where *D. suweonensis* occurred. Data on protected areas were downloaded from the Protected Planet database, set by the *IUCN & UNEP-WCMC (2016)*. We subsequently noted the number of sites within any protected area, as well as "sites that do not meet the standard definition of a protected
area but do achieve conservation in the long-term under national and international agreements'' (*IUCN & UNEP-WCMC, 2016*).

## Origin of agricultural flood waters

To analyse the impact of agricultural flood water on *D. suweonensis* distribution, we asked rice farmers for the origin of the water used to flood their rice paddies. This survey was restricted to the general riverine basin surrounding the city of Iksan, south of the Geum River. To be included in the analysis, the origin of the water for a rice paddy complex had to be confirmed by at least two different farmers (Fig. 2). Data collection was limited to sites where surveys for *D. suweonensis* were conducted. The area surveyed south of the city of Gunsan and the Mankyeong River had to be excluded from the analysis due to lack of traceability of the origin of agricultural flood water (Fig. 2).

## Data analysis and optimal conservation sites

For subsequent analyses, we binary encoded the presence of the species, the presence of greenhouses and the presence of permanent human infrastructures within the rice-paddy complexes. We first determined the range of the species, based on presence data points (Fig. 1). We defined the potential range of the species based on the non-interruption of landscape variables that are within the range used by the species. We also delineated the ancestral range of the species, defined as the potential range of the species before human development. Namely, a site was considered potential for the species if <120 m of altitude and within the same water basin as a known population, excluding cities and urban area >1 km$^2$ (Fig. 1).

We then defined the overlap between species range and reclaimed area to estimate the land use by the species, and calculated the overlap between species range and protected areas. Descriptive statistics were used to characterise the impact of these landscape variables in both cases.

We hypothesised the origin of the water to be important if linked to the Geum River. This geographic area was chosen due to the clear segregation between areas flooded with water from different origins. We indexed the occurrence of *D. suweonensis* at the sites surveyed in relation to the binary encoding of the origin of flood water, from the Geum River. We subsequently assessed whether distribution of *D. suweonensis* was random in relation to agricultural flood water.

Finally, we developed a plan for an optimal site for the protection of the species. From survey presence data, we calculated averages for water quality (pH and conductivity) as proxies for a larger set of values important for the species (A Borzée et al., 2014, unpublished data), the continuity with rivers and forests, and the sphericity of sites. For sites surveyed over multiple years, the abiotic variables used for the calculation of the species' preferences were restricted to the latest data point. This choice to restrict the analysis to the survey presence data followed recent documented local extinctions, and the potential for other undocumented local extinction due to water quality, salinity, competition and land-use among others, and because these variables are important to ecological preferences of species. All analyses were conducted with SPSS (v. 21.0, SPSS, Inc., Chicago, IL, USA),

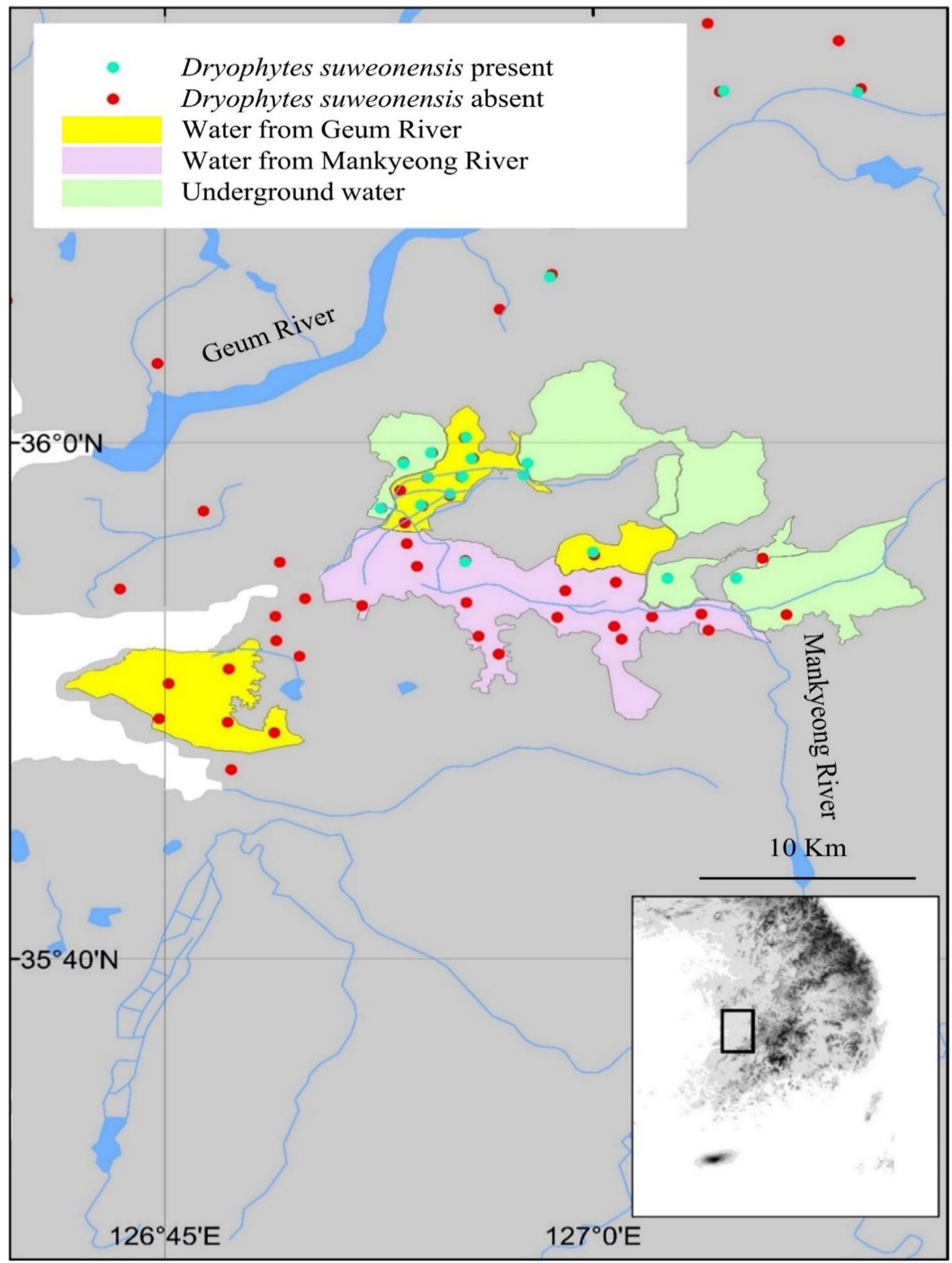

**Figure 2 Relationship between flood water origin and species presence.** Most of the flood water matching with the occurrence of *Dryophytes suweonensis* originated from the Geum River (53.3%), followed by underground water (40%), while the remaining 6.7% of sites were flooded by the Mankyeong river. This analysis is restricted to the area shown on the map.

and maps were generated with ArcMap 9.3 (Environmental Systems Resource Institute, Redlands, California, USA).

## RESULTS

During the surveys conducted in 2014, only 358 sites out of the 789 sites pre-selected were potentially habitable for the species as urban development and agricultural conversion eliminated the 431 remaining sites. That is, these sites were beyond the ecological requirements of the species as there was no standing water; instead the sites were mostly greenhouses, apartment complexes or dry crops. Within the 358 habitable sites, we found calling *Dryophytes suweonensis* at 100 sites, while the species was not detected at 258 sites. In 2015, calling *D. suweonensis* were detected at 106 sites total, from 94 of the 100 sites where the species was detected in 2014, and 12 of the new sites. In 2016, the species was detected at 109 sites total, 94 of the 2014 sites, 12 of the 2015 sites and three of the new sites. The 94 sites originating from the 2014 dataset where the species was detected in 2015 and 2016 were the same. The species was not detected at the five remaining sites where it had been found in 2014. The 12 sites where the species was detected in 2015 were included in the surveys in 2016, and the species was again detected at all 12 sites. For all subsequent analyses, we assess the species to be present at the 114 sites where the species was detected at least once. This includes the 113 sites surveyed over three years and the site behind the CCZ and these sites are distributed over *circa* 4,300 km$^2$ (Fig. 1). However, this species is under significant threat of local extinction at the five sites where the species was detected in 2014 only; a new motorway was built during the study period in the Bay of Asan (Fig. 1).

### Range, ancestral range and current optimal range

The southern boundary of *D. suweonensis*' distribution was extended 120 km southwards from the previous assessment (*Kim et al., 2012*). The distribution of *D. suweonensis* ranges from the southern banks of the Imjin River to the northern banks of the Mankyeong River, on a 220 km north-south transect. The range of the species spans 95 km longitudinally, with the westernmost known population in Hongseong area and the easternmost in Wonju (Fig. 1).

The potential range of the species, defined as the area where ecological preferences of the species are matched, is situated at the same latitude as the one where the species was detected, but extends 25 km further west from the westernmost site where the species was detected towards the reclaimed Cheonsu bay. In addition, the corridor of low lands between Nonsan, Gongju and Cheongju matches the habitat required for the species, but no surveys were conducted in that area, as primarily estimated to be too far and disconnected from the range of the species to be a potential breeding area. When compared with the potential range of the species before human development, referred here as ancestral range, the land surface area usable by the species decreased by 729 km$^2$ (Fig. 1).

### Overlap between reclaimed lands and protected area

Out of the 114 sites where *D. suweonensis* was detected, a total of 30 sites were enlarged and 15 sites were created through land reclamation. The remaining 69 sites were not

**Table 1  Descriptive statistics for abiotic variables of interest collected from all sites where *Dryophytes suweonensis* was present.**

|  | N | Min | Max | Mean | Std |
|---|---|---|---|---|---|
| Water pH | 114 | 7.20 | 10.20 | 8.32 | 0.32 |
| Water conductivity (µS) | 114 | 83.50 | 5720.00 | 792.19 | 740.47 |
| Surface area (m²) | 114 | 0.31 | 26.09 | 4.78 | 4.36 |
| Max. length (km) | 114 | 1.10 | 301.00 | 6.30 | 27.89 |
| Continuity with forests (km) | 114 | 0.00 | 14.10 | 3.87 | 2.83 |
| Continuity with rivers (km) | 114 | 0.00 | 9.20 | 1.17 | 1.79 |
| Sphericity | 114 | 0.01 | 2.87 | 1.15 | 0.65 |

impacted by land reclamation. When combining all sites impacted by land reclamation, they represent 39.47% of the sites where *D. suweonensis* was present. When focusing on the overlap between the occurrence of *D. suweonensis* and protected areas, only a single site was selected, South of Pyeongtaek, protected under "Water Source Protection Area". In this protected site, only the riverine system at the edge of the site is protected, putatively used by *D. suweonensis* for hibernation and not for breeding.

## Origin of agricultural flood waters

This analysis is based on a subset of sites in the southern distribution of the species (Fig. 2). A total of 53.3% of sites where *D. suweonensis* was present overlapped with agricultural floods originating from the Geum River (Fig. 2), highlighting the non-random occurrence of the species in this area. Few sites surveyed in the putatively suitable areas using water from the Mankyeong River had *D. suweonensis* (1/15 sites; 7%), while those in areas using water from the Geum River had higher presence (8/17 sites; 49%) and an even high proportion of sites showed presence in the areas utilising underground water (6/8 sites; 75%). Autocorrelation of the origin of flood water is likely, although of minor importance in this study and unlikely to impact the result of the statistical analyses.

## Assessment of optimal conservation site

The environmental variables for *D. suweonensis* (Table 1) showed an average pH of 8.32 and average conductivity of 792.19 µS. The average sphericity was 1.15, meaning that sites were more round than elongated in general. The majority of sites where *D. suweonensis* occurred had permanent man-made infrastructures (52.9%) and temporary structures (i.e., greenhouses, 68.9%) within the rice-paddy complexes.

Depiction of the sites adequate for the conservation of *D. suweonensis* (Fig. 3) was supplemented by vegetation lists from *Borzée & Jang (2015)*, and landscape information matching the current habitat of *D. suweonensis*. Rice paddies are delimited by levees roughly 40 cm wide and 20 to 60 cm high, covered with grasses, and used by treefrogs for basking, foraging, and sheltering (*Borzée et al., 2016*). The overhead view of the designed site highlights the need for continuity with forests and rivers to match the preferences of the species (Fig. 3A), while the lateral view (Fig. 3B) describes depth and vegetation characteristics required for the species.

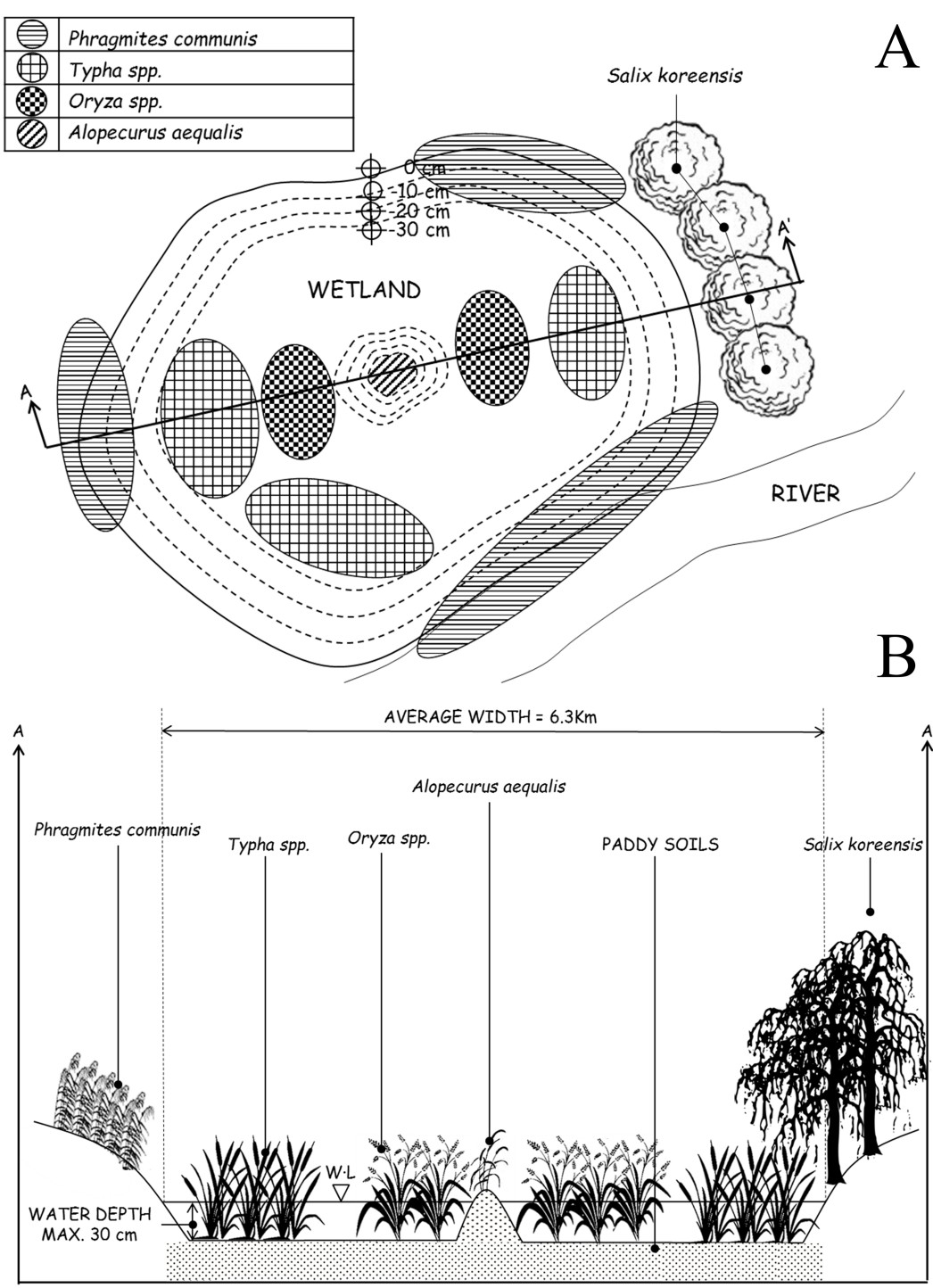

**Figure 3** (A) Overhead view of the site optimally designed to follow ecological preferences demonstrated by *Dryophytes suweonensis*. The cut AA' is reported in **Fig. 3B**. The figure is not to scale. Water depths indicated are matching with the average depth of rice paddies, and therefore acceptable if not optimal for the species, and vegetation data is extracted from *Borzée & Jang (2015)*. (B) Lateral view of the site optimally designed to follow ecological preferences demonstrated by *Dryophytes suweonensis*. Water level (WL) originates from the only known natural site with *Dryophytes suweonensis* (*Borzée & Jang, 2015*).

## DISCUSSION

This study highlights the importance of analysing data on the presence/absence and habitat characteristics of species for their conservation. The known range of *Dryophytes suweonensis* was doubled by the data collected over this three years study, highlighting the need for a different approach to the selection of sites for the conservation of the species. These new data show that the increase in known range is due to the inclusion of a large number of sites in reclaimed area from post-war agricultural governmental development. The apparent expansion of the species' known range is, however, countered by several potential local extirpations, such as all the sites in the area of Suweon where the holotype for *D. suweonensis* was described (*Kuramoto, 1980*; *Park, Jeong & Jang, 2013*).

The species still matches the criteria B1ab(i,ii,iii,iv) for listing as "endangered" under the criteria of the International Union for Conservation of nature (IUCN) red list of endangered species. It has an extent of occurrence <5,000 km$^2$, a severely fragmented population with a continuing observed decline for extent of occurrence, area of occupancy, quality of habitat and the number of locations or subpopulations. At present, the protection of *D. suweonensis* is not ensured because no populations are located within a protected area. Only the edges of a single site are overlapping with a protected area, south of Pyeongtaek. A single protected site is inadequate for the conservation of an endangered species.

The description of the potential range for *D. suweonensis* shows that an area around Cheongju may be adequate for the species to thrive. However, that area was not included in the initial surveys, due to the lack of knowledge of such a potential wide range for the species. Similarly, sites such as Baengnyeong or Seogmo Islands may be suitable but could not be accessed due to their limited access to non-military personnel. Another potential significant range increase would be within the Democratic People's Republic of Korea, as the species is known to occur around Pyongyang (*Chun et al., 2012*).

Encroachment on the species' range by development (431 sites), such as at the sites around the city of Suweon, has been partially counter-balanced by the land reclamation projects for rice agriculture (15 sites) implemented at a very large scale in the Republic of Korea during the second half of the last century. The presence of *D. suweonensis* on reclaimed land shows that the species possesses the potential for dispersal despite a lower dispersal ability than the sympatric *Dryophytes japonicus* (*Borzée & Jang, 2016*). This shift in range is thus linked to rice cultivation and may have been an on-going process since early human agriculture *circa* 5000 years ago (*Fuller, Harvey & Qin, 2007*; *Fuller, Qin & Harvey, 2008*).

Furthermore, numerous *D. suweonensis* populations are isolated from each other, with urbanization resulting in multiple landscape barriers within and among potential metapopulations. This calls for a long-term study of population dynamics and network analysis for the species. We would expect the population to be larger at reclaimed sites, due to lower levels of encroachment and fragmentation.

The water origin analysis showed that frogs occur at sites flooded by water originating from the Geum River and underground water sources. However, the species was not detected at sites flooded by water originating from the Mankyeong River. Thus, water

originating from underground water bodies and pumped to the surface for agricultural purposes may be adequate for *D. suweonensis*. The areas flooded by river water may be the ones that were seasonally flooded before landscape modifications by humans, and the absence of *D. suweonensis* at some sites could relate to water quality. This idea is potentially supported by the absence of individuals at the only site flooded by water originating from the Geum River south of the Mankyeong River. However, as the water is brought by aerial channels, it is possible that some individuals *D. suweonensis* will drift south to this area in the future and establish new colonies, or perhaps hybridise with the *D. japonicus* present at the site (*Borzée, Fong & Jang, 2015*).

Conservation of a species often requires the restoration of the species' habitat (*Rannap, Lohmus & Briggs, 2009*). The design of an optimal site for the protection of *D. suweonensis* highlights the need for very large continuous flood plains. However, such large plains are becoming frequently uncommon because of encroachment, and management plans have to be set before these sites disappear. Furthermore, the presence of bullfrogs in the southern part of the range, may have a known negative impact on the species (*Borzée et al., 2017*), and demonstrates that the first steps of conservation management plans for the species have to be conducted at any site where the species would be protected/re-introduced. Finally, as the species is still present across a range similar to its ancestral range, we do not recommend ex-situ conservation projects at this time, nor translocations to new sites that would be outside of the ancestral range for the species.

## ACKNOWLEDGEMENTS

We are extremely thankful to Mr. Yu Sang Hong for all the communication with the farmers, and without whom a large part of this manuscript would not exist.

### Funding

This project was supported by three Small Grants for Science and Conservation in 2014, 2015 and 2016 from The Biodiversity Foundation to Amaël Borzée. The project was also funded by a Research Grant from the National Research Foundation of Korea (#2017R1A2B2003579) and a research Grant from National Geographic Asia (# 2-2016-1632-001-1) to Yikweon Jang. There was no additional external funding received for this study. The funders had no role in study design, data collection and analysis, decision to publish, or preparation of the manuscript.

### Grant Disclosures

The following grant information was disclosed by the authors:
The Biodiversity Foundation.
National Research Foundation of Korea: #2017R1A2B2003579.
National Geographic Asia: # 2-2016-1632-001-1.

## Competing Interests

The authors declare there are no competing interests.

## Author Contributions

- Amaël Borzée conceived and designed the experiments, performed the experiments, analyzed the data, contributed reagents/materials/analysis tools, wrote the paper, prepared figures and/or tables.
- Kyungmin Kim performed the experiments, analyzed the data.
- Kyongman Heo performed the experiments.
- Piotr G. Jablonski reviewed drafts of the paper.
- Yikweon Jang conceived and designed the experiments, analyzed the data, contributed reagents/materials/analysis tools, prepared figures and/or tables, reviewed drafts of the paper, supervision.

## Animal Ethics

The following information was supplied relating to ethical approvals (i.e., approving body and any reference numbers):

The work conducted here did not require approvals as it only includes call surveys. No individual was handled during this study.

## Data Availability

The raw data has been uploaded as a Supplemental File.

## Supplemental Information

Supplemental information for this article can be found online at http://dx.doi.org/10.7717/peerj.3872#supplemental-information.

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
