# Peer review of "Impact of land reclamation and agricultural water regime on the distribution and conservation status of the endangered Dryophytes suweonensis"

_PeerJ, doi:10.7717/peerj.3872_

## Round 0.1 · original submission · Minor Revisions

· Academic Editor

Minor Revisions

This manuscript describes a revision to the known distribution of Dryophytes suweonensis. All three reviewers have agreed that this study is interesting and a valuable contribution to the literature. Most of their comments and those listed below are editorial in nature and will greatly improve the readability of the manuscript. Please consider these suggestions and amend the document accordingly. Additionally, several questions have been raised about statistical issues that should be addressed. R2 raised questions about the survey methodology that should be answered.

In addition, please clarify why the area in Figure 2 was chosen to examine the effects of water source. Also, please provide a clear hypothesis in the aims section (later in the manuscript, you do say you hypothesised this area would be important but you don’t say why!). As is, it is unclear why you chose to do this but it is obvious that there is clearly a difference in the effects of these two rivers. Presumably there was a reason that this area was targeted? Expanding on this will improve this manuscript.

Line 23 Suggest changing “potential” to “potentially suitable”

Line 24 The “entire range” seems incorrect here – later you state that they are also found in North Korea. Please clarify.

Lines 24-26 Suggest, “We then assessed whether D. Suweonensis was found in the current and ancestral predicted ranges, reclaimed and protected areas, and how the presence of agricultural floodwater affected its occurence.”

Line 35 Change “site” to “sites”

Line 38 Suggest “and most are likely”

Line 40 Remove comma after “urbanization”

Line 48 Suggest “Knowledge of species’ habitat preferences provides”

Line 56 Remove comma after “number”

Line 58 Please explain what is meant by “debated effects”

Line 138 Suggest “beds, which have been converted”

Line 156 Suggest “To be included in the analysis,”

Line 165 Suggest “We first determined the range”

Line 72 Remove comma after “valleys”

Lines 78-83 I agree with R3 that your aims should be expanded to more closely describe what you have done.

Line 174-177 Explain why the area you used was chosen.

Line 178 Suggest “Finally, we developed a plan for an optimal”

Line 194 Change to “calling D. suweonensis”

Line 206 Suggest “ancestral range” instead of “ancestral potential range” – you do go on to define this term so I think you could keep it simple here to avoid confusion (ancestral suggests past whereas potential suggests future).

Line 216 Please say why surveys were not conducted in this area – what caused them to be excluded?

Lines 229-235 Please explain that this analysis refers on to the restricted area included in Fig 2.

Line 232 Add “in this area” after “species”

Line 232-234 “The second most common” is funny here since you only looked at two sources. To what does the 20% refer? From looking at Fig 2, it appears that only 1 of 15 sites flooded by the Mankyeong River contained the study species. Can you please clarify this section.

Line 254 Change to “range of D. suweonensis”

Line 271 Please explain what knowledge was lacking that led to the exclusion of this area.

Line 272 Insert “may be suitable but” between “Islands” and “could”

Line 279 Spell out “Dryophytes japonicas”

Line 284 Change “study on” to “study of”

Line 296 Change “at least” to “perhaps”

Line 299 Remove “is unusual as it”

Lines 301-302 Change “the disappearance of such sites” to “these sites disappear”.

Lines 302-303 Suggest “range, which are known to have negative impacts”

Line 304 Please clarify what “preliminary work for the protection” means.

Line 306 Insert “at this time” after “projects”

Line 442 Remove “potential” after “ancestral”

Figure 1 Add scale bar (kilometres) and indicate the location of the single protected site where the species occurs.

Figure 2 legend Please clarify that this information refers to the restricted area included in the water source analysis and not to the whole study area.

Reviewer 1 ·

Basic reporting

Seems very good. Relatively good English with a few small errors or stylistic problems as noted in the General Comments section. Well structured in standard format.

Experimental design

Seems fine to address most of the questions asked. Probably not adequate for some, though--extreme spatial autocorrelation of sources of water means that many other grographical variables may be confounded with water source; see comments on Figure 2 in general comments section and comment on lines 229-235 in "Validity of the Findings"

Validity of the findings

229-235 Not sure I see the significance of this? I assume that the percentages of sources of water just relate to the species’ geographic distribution. Sources of floodwaters are highly spatially autocorrelated and cannot be regarded as independent for the pruposes of things like chi-squared tests.

See also Figure 2 comment in general comments section and comment on 287-297

Additional comments

Most of these are simple English comments, however a few are more substantive.

54 should either be “a species’ range and habitat preferences” or “species’ ranges and habitat preferences”

60 “The amphibia class -> “The Class Amphibia”

64 “Besides, those farmlads” -> “Farmlands”

76 “lead to expect” -> “led us to expect” lead is a metal, or something one does to guide someone to a place (pronounced "leed"), led is the past tense of the guiding meaning. Good old English...

79 “karyotype warranting” -> “karyotype, warranting”

88 “negative” -> “negatives”

100 “absence for” -> “absence of”

110 “study for” -> “study of”

139 “presence of the” -> “presence of”

174 needs a rewrite. Do you mean you hypothesized that it was important that water be linked to the Gxxx river? I say Gxxx because here you refer to the Geum river and above (156) you referred to the Gum river, and I am on an aeroplane with no access to a map to check which is correct.

176-177 “We subsequently assessed the random distribution of D. suweonensis in relation of the agricultural flood water.” -> “We subsequently assessed whether
distribution of D. suweonensis was random in relation to that of agricultural flood water.”

184 “and variables such as water quality are important the” -> and because variables such as water quality are important in the”

195 “were” -> “where”

202-203 “surveyed three” -> “surveyed over three”

218 This is important—did the range decrease by 729 km2 or from 729 km2 to some lesser area? Just a typo I’m sure

241 “temporal” -> “temporary” if that is what you mean; temporal is definitely wrong, anyway. Dr. Who’s TARDIS is a temporal structure; I don’t know of any others.

254-255 “The known range of Dryophytes suweonensis has doubled over the three years of this study” -> “The known range of Dryophytes suweonensis was doubled by the data collected over the three years of this study”

258-259 “This new knowledge is, however, sobered down by several potential local extirpations” -> “The apparent expansion of the species’ known range is, however, countered by several potential local extirpations”

260 “”Besides, the” -> “The”

262 “Namely, because of an extent” -> “It has an extent”

265 “Besides, the” -> “At present, the”

267 “It is however” -> “This is”

272 “Accordingly” -> “Additionally”

276 “Encroachment had been” -> “Encroachment on the species’ range by development has been”

277 “carried at” -> “carried out at”

283 if there are numerous major landscape barriers within potential metapopulations, they probably are not single metapopulations, but either groupings of isolated populations or groupings of isolated metapopulations. This needs to be revised. Just adding the word “potential” preceding “metapopulations” would do the trick.

287-297 See my comment in the “Validity of the Findings” section about this. If this is a real thing it needs to be explained more thoroughly and justified better. Otherwise it may just be the result of any of the many factors that might be restricting geographic distribution of the species.

306-307 “neither than introduction at new sites” is completely wrong wording; I really am not sure what you mean; do you mean you do not recommend ex-situ conservation other than introduction at new sites outside the present range (commonly called translocation), or do you mean that you recommend neither ex-situ conservation nor translocation?

Figure 1 legend I would suggest saying
(blue dot) Dryophytes suweonensis present
(red dot) Dryophytes suweonensis absent

i.e. reverse the wording for the present and absent categories. Not really necessary but clearer and better English

449 “was” -> “were”

Figure 2—note the extreme degree of spatial autocorrelation, i.e. each point sampled is very non-ndeendent from many others in terms of the source of water, so that regarding them as independent and conducting a chi-squared test, which assumes that, is incorrect.

452 “Bird view” -> “Bird’s-eye view” or “Overhead view”

457 “Lateral view the” -> “Lateral view of the”

·

Basic reporting

I found this to be a very well written paper, based on several years of survey data. Clearly a lot of work was done to gather the data. I make some minor suggestions below to improve the grammar.

Line 28: replace “evidences” with “evidence”

Line 30: replace “Besides,” with “In addition,”

Line 33: delete “ultimately” as this word is not needed in the sentence

Line 41: I don’t know if you mean “extinction risks” (plural) or “extinction risk”. You should say either “extinction risks depend on” or “extinction risk depends on””

Line 44-45: Change so it reads “The lack of knowledge of the distribution of a species has already resulted in extinction that could have been easily avoided”.

Line 45: Change to “For example, the Tecopa pupfish….”.

Line 48: Change “Knowledge on” to “Knowledge of”

Line 50: Say “subspecies of Ursini’s viper”, not “subspecies of the Ursini’s viper”

Line 51: Change to “was known to occur only in Greece and at a single locality in Albania.”

Line 57: Change to “The occurrence of a species ….”

Line 58-59: you finish the sentence with “and despite the low occurrence of endangered species within protected areas (Brooks et al. 2004).” I don’t understand what you mean by this, in relation to the rest of the sentence. Can you clarify?

Line 60: Change “amphibian class” to “Class Amphibia”

Line 64: Change “Besides” to “Furthermore”

Line 66: Say “where rice production has decreased …”

Line 72: Change “occur on” to “occur in”

Line 76-77: Change to “lead to the expectation of a broader distribution for the species”

Line 79: Change “enterprised” to “aimed”

Line 99: this is the first time you mention D. japonicus. You should say something about why this species is important to your study.

Line 110: Change to “occurrence of this species”

Line 116: Change “before” to “adjacent to”

Line 125-126: Do you mean you had previously measured the detection range of frog calls, or you did this at the end of each site survey. I suspect the former, but please clarify.

Line 160: Change “traceability in the origin” to “traceability of the origin”

Line 171: Insert a comma after “species”

Line 177: Change “in relation of” to “in relation to”

Line 179: Change “From presence data from the surveys” to “From survey presence data”

Line 182: Say either “the species’ preferences” or delete “the” and just say “species preferences”

Line 184: Change “the ecological preferences” to “ecological preferences”

Line 191 to 193: I would make this sentence more concise by saying something like “That is, the sites were beyond the ecological requirements of the species as there
was no standing water; instead the sites were mostly greenhouses, apartment complexes or dry crops.” If that is what you mean.

Line 195-196: Change “from 94 of the 100 sites were the species was detected” to “from 94 of the 100 sites where the species was detected”

Line 198: delete the comma in the sentence as it is not necessary.

Line 204” I’m not sure “aggravated threats” is the best phrase. Perhaps change to “significant threats” or similar.

Line 215: Change “Besides” to “In addition”

Line 218: you say “the species decreased by from 729 km”. It should either be “by 729 km” if that is what you mean, or “from 729 km to XXXX” if that is what you mean. But don’t say “by from” in the same sentence.

Line 247: I would change “bird-eye view” to “aerial view”

Line 258: Change “sobered down” to “tempered by” or similar

Line 260: Replace “Besides” with “Furthermore”

Line 262-265: The sentence “Namely, because of an extent of occurrence < 5000 km2, a severely fragmented population, with a continuing decline observed, estimated, inferred or projected for extent of occurrence; area of occupancy; area, extent and/or quality of habitat; and the number of locations or subpopulations” is a bit confusing. I am not sure exactly what you mean, but this should be clarified. For example, what does “and the number of locations or subpopulations” refer to? Is that referring to a previous part of the sentence regarding continuing declines (observed, estimated, inferred, projected), or are you just talking about the actual number of locations and subpopulations being the issue?

Line 270: Change “strive” to “exist”

Line 271: Change “knowledge on” to “knowledge of”

Line 277: Change “carried at” to “implemented at”

Line 302: Change “southern part on” to “southern part of”

Line 306: Change “neither than” to “nor”

Line 447: You need a better Figure heading. Keep what you have, but start with something like “Relationship between water type and species presence”.

Experimental design

My main question relates to the use of a single 5 minute survey each year to assess presence or absence of the frog species at a site. This may be justifiable - I just don't know. See my comments below in the Validity of the Findings Section.

Validity of the findings

I have some questions about the data analysis. The authors detected frogs at 114 sites, and never found frogs at something like another 250 sites. Each site was surveyed during a single 5 minute survey (listening for frog calls) during the breeding period. My first question relates to detection probability. How confident are the authors that a single 5 minute survey is sufficient to detect the presence of the species at a site? There is a large literature on this for amphibians in general. For some species, detection probability is quite high (close to 1; thus a single survey is an adequate sampling period), but for other species it is low and so multiple surveys in a short time frame are required to be confident of species presence or absence. This obviously can have significant implications for identifying the distribution range of the species, as well as identifying habitat correlates. Do the authors have any data on the detection probability of their target frog species? If so, and it is high, that would help justify the use of a single 5 minute survey at a site once a year. I agree that the authors have expanded the range of the species, I’m just not sure if the actual range is even larger than they suspect – if the species has low detection probability.

As I understand it, to identify important habitat variables for species presence, the authors focused on the 114 sites where the species was present, and did not include data from the other 250 sites (approximately) where the species was never detected. Is that correct? If yes, I don’t understand why the authors didn’t use the data for all 350+ sites (species detected or not detected) in an analysis like logistic regression to see which of the habitat variables was significant in explaining the detection or non-detection of the species at sites? Unless you think detection probability is low, and therefore sites where the species was not detected are too questionable to use to identify habitat correlates. Or there were other reasons to not expect the presence of frogs at some sites independent of local habitat conditions. There may be a reason for their approach to analysis, but the authors should clarify their decision. Or if I have misunderstood the analysis, the authors should be more explicit in describing the analysis they used.

Additional comments

No additional comments.

·

Basic reporting

Nicely conceived maps and figure. I would like to see a few clarifications of them as specified in the manuscript enclosed. I was confused by your use of ancestral and current distribution and your suggestion that rice seedlings are necessary for the species. If this is true ancestral has a very restricted meaning.

Experimental design

I think your a priori selection of field sites to survey is excellent, but you need to highlight how excellent it is by telling us more about how you went about drawing the boundaries on the original area that you are focused on and how you the selected the individual sites within this larger area.

Validity of the findings

I suggest you develop your figure 3 more clearly. It is based on other work in large part (e.g. vegetation characteristics) so it kind of drops out of the blue. But, it is a beautiful figure.

Additional comments

This is an important study that has many good features associated with it. Primarily the extensive field survey work which documents the occurrence of this endangered species and data associated with positive impacts of both rice paddy development and negative impacts of urban encroachment. A particularly strong component of the study is the large effort that went into gathering presence absence data and correlating this with human land use modifications. The paper is difficult to read and I have made many suggestions, corrections, and comments on the enclosed manuscript. I edited in MSword so the track change comments came across with MOU and REV associated with them. They have not meaning. They are all my comments.

---

## Round 0.2 · Minor Revisions

· Academic Editor

Minor Revisions

Thank you for the care you have taken to address the previous round of comments. The manuscript is substantially improved. However, there are a few remaining issues that need to be addressed before this manuscript can move forward.

Line 22 Please change “ancestral predicted” to “predicted ancestral”

Line 42 Suggest changing to “Lack of knowledge of species’ distributions has already resulted in extinctions that…”

Line 55 Please change “number” to “numbers”

Lines 77-84 Please add to the paragraph that you also estimated the impact of urbanization on the species range (ancestral range lost).

Line 89 It is still unclear whether this species ONLY uses rice seedlings as supports when calling and the reference doesn’t state this. Please clarify if this is the case and how this is known.

Line 90 Please add full stop at end of sentence.

Line 91 Please change “lead” to “led”

Lines 115-116 Suggest changing “until ensuring the absence of the species at the southernmost locations surveyed” to “until reaching a point past where the species was no longer detected”

Lines 135-136 Suggest inserting “determining” between “for” and “the” and suggest changing “species to be present as” to “species’ presence because”. Also, this sentence needs a reference.

Lines 172-173 Suggest you make this two sentences: “We delineated the potential range of the species based on the non-interruption of landscape variables that are within the range used by the species. We also delineated the ancestral range of the species (defined as the potential range of the species before human development).” This will clarify these terms at first mention.

Lines 211-213 Suggest changing to: “…CCZ and these sites are distributed over circa 4300km2 (Fig. 1). However, this species is under significant threat of local extinction at the five sites where the species was detected only in 2014; a new motorway was built in this area during the study period (Fig. 1, Bay of Asan).”

Line 223 Please change “bay” to “Bay”

Line 225 Suggest changing to “…area because it was initially estimated to be too far away and disconnected…”

Lines 239-246 This section is still problematic. It remains unclear what the goal of this analysis is. First, presenting occurrence information (as you have done in the current version of the manuscript) without correcting this for the total number of suitable sites within each area of water origin isn’t really meaningful. Isn’t the proportion of presence vs. absence sites per water origin what’s really interesting here? For example, few sites surveyed in the putatively suitable areas using water from the Mankyeong River had frogs (1/15 sites; 7%), while those in areas using Geum water had higher presence (8/17 sites; 49%) and an even high proportion of sites showed presence in the area utilising underground water (6/8 sites; 75%)? If this is the case, please change the paragraph to reflect this. Otherwise, please clarify.

Line 248 Suggest changing this section title to “Assessment of optimal conservation site”

Line 263 Suggest inserting “analyzing” between “of” and “data”

Line 265 Suggesting changing to “…three year study, highlighting the need…”

Lines 267-268 Suggest changing to “…is due to the inclusion of a large number of sites in reclaimed areas from post-war agricultural government development. The apparent…”

Line 277 Please change “is” to “are”

Lines 278-279 Suggest changing to “…Pyeongtaek. A single protected site is inadequate for the conservation of an endangered species.”

Lines 287-290 Suggest adding a comment in parentheses here that 431 sites have been lost and only 15 gained, in line with previous comments from R3 about this sentence.

Line 296 Please remove “between and”. “Between” is normally used in a comparison of two, while “among” is used to compare multiple items.

Lines 300-311 This interpretation of your results does not seem to make sense. How can you say that frogs are not found in areas where water originates from underground water bodies when hardly any of these areas were included in your survey (n=8, over a large area depicted in Fig 2) and those that were included had high presence (6/8 sites)? If anything, what Fig 2 seems to indicate is that the Mankyeong River flooded areas are not suitable for this species. Can you please clarify this?

Line 317 Suggest changing “shows” to “and demonstrates”

Line 319 Suggest changing “on” to “across”

Lines 344-345 Please change the title of this reference to reflect the published version.

Figure 1 Please change caption to read “Potential current range” and “Potential ancestral range”

Figure 3b Reviewer 3’s previous comments about this figure were simply suggesting that you make a mirror image of 3b so that, for example, the Salix kareensis is on the right side of the image as it is in Figure 3a. This makes your concept of the ideal site more obvious to your reader. Can this be done please?

---

## Round 0.3 · accepted · Accept

· Academic Editor

Accept

Thank you for your diligence in addressing editorial and reviewer comments.